# Mechanochemical Synthesis and Structure of the Tetrahydrate and Mesoporous Anhydrous Metforminium(2+)-*N*,*N*′-1,4-Phenylenedioxalamic Acid (1:2) Salt: The Role of Hydrogen Bonding and n→π * Charge Assisted Interactions

**DOI:** 10.3390/pharmaceutics12100998

**Published:** 2020-10-21

**Authors:** Sayuri Chong-Canto, Efrén V. García-Báez, Francisco J. Martínez-Martínez, Angel A. Ramos-Organillo, Itzia I. Padilla-Martínez

**Affiliations:** 1Laboratorio de Química Supramolecular y Nanociencias, Instituto Politécnico Nacional-UPIBI, Av. Acueducto s/n Barrio la Laguna Ticomán, Ciudad de México C.P. 07340, Mexico; schongc0800@alumno.ipn.mx (S.C.-C.); egarciaba@ipn.mx (E.V.G.-B.); 2Facultad de Ciencias Químicas, Universidad de Colima, Km. 9 Carretera Colima-Coquimatlán, Coquimatlán C.P. 28400, Colima, Mexico; fjmartin@ucol.mx (F.J.M.-M.); aaramos@ucol.mx (A.A.R.-O.)

**Keywords:** metformin cocrystal, mechanochemical synthesis, dicationic metformin, water channels, pi-interactions, mesoporous anhydrate

## Abstract

A new organic salt of metformin, an antidiabetic drug, and *N*,*N*′-(1,4-phenylene)dioxalamic acid, was mechanochemically synthesized, purified by crystallization from solution and characterized by single X-ray crystallography. The structure revealed a salt-type crystal hydrate composed of one dicationic metformin unit, two monoanionic units of the acid and four water molecules, namely H_2_Mf(HpOXA)_2_∙4H_2_O. X-ray powder, IR, ^13^C-CPMAS, thermal and BET adsorption–desorption analyses were performed to elucidate the structure of the molecular and supramolecular structure of the anhydrous microcrystalline mesoporous solid H_2_Mf(HpOXA)_2_. The results suggest that their structures, conformation and hydrogen bonding schemes are very similar. To the best of our knowledge, the selective formation of the monoanion HpOXA^−^, as well as its structure in the solid, is herein reported for the first time. Regular O(δ−)∙∙∙C(δ), O(δ−)∙∙∙N^+^ and bifacial O(δ−)∙∙∙C(δ)∙∙∙O(δ−) of n→π * charge-assisted interactions are herein described in H_2_MfA organic salts which could be responsible of the interactions of metformin in biologic systems. The results support the participation of n→π * charge-assisted interactions independently, and not just as a short contact imposed by the geometric constraint due to the hydrogen bonding patterns.

## 1. Introduction

Cocrystallization has become a growing discipline of interest in pharmaceutical sciences, since its application has been demonstrated to modify the physical-chemical properties of known drugs [1]. Metformin–HCl (HMfCl) is an oral anti-hyperglycemic drug, used worldwide for the treatment of non-insulin-dependent diabetes mellitus. It improves glucose tolerance, lowering plasma glucose levels and glycated hemoglobin, particularly in overweight and obese patients. Metformin is one of the most frequently used small molecules for preparing combined drugs such as Segluromet (metformin–ertugliflozin), Metaglip (metformin–glipizide) and Glucovance (metformin–glyburide). These are examples of the fourteen combinations approved by FDA [2], some of which have been characterized by monocrystal X-ray diffraction such as metformin–glimepiride [3] and metformin–salicylic acid [4]. Due to its pharmaceutical importance, there is a great interest in metformin cocrystals, which has led to several patents related to the formation of organic salts, compiled elsewhere [5].

In addition, the structure and electronic structural details of metformin and its cocrystals have recently been summarized, in the context of biguanide compounds [6]. Metformin is a privileged small molecule; it possesses conformational flexibility and is capable of forming strong hydrogen bonding interactions, which will determine its structure and activity in biological processes. Metformin hydrochloride is known to crystallize into two conformational polymorphs, namely, A [7] and B [8]. The Me_2_N–C–N–C backbone adopts the U conformation in the thermodynamic phase A (Me_2_NCNC torsion angle value of 53.7°), whereas the form of an S backbone is observed in the metastable polymorph B (Me_2_NCNC torsion angle value of 129.1°). The calculated minima are near 50° and 160°, respectively [9].

On the other hand, oxalamic or oxamic acids are the amide-carboxylic acid derivatives of the oxalic acid. They have been recognized because of their high potential in crystal engineering and molecular recognition due to their bifunctionality [10]. As far as *N,N*′-(1,4-phenylene)dioxalamic acid (H_2_pOXA) and its 1,3-isomer (H_2_mOXA) are concerned, they were first reported in the mid 1970s [11,12] as antiallergic agents as well as anti-inflammatory compounds [13]. More recently, both have attracted interest as coordinating ligands for metals [14,15]. Theoretical studies and experimental work [16] have demonstrated the high flexibility of the oxalic acid derivatives. Both oxalyl carbonyls are usually in *anti* disposition [17], but they less frequently adopt the *syn* disposition, induced by steric constraints [18,19] or by coordination with metals in the form of carboxylates [14].

Finally, mechanochemical synthesis consists of the application of mechanical energy to induce a chemical reaction, its use and applications have been recently reviewed [20]. Liquid-assisted grinding (LAG) makes use of some drops of solvent to provide greater molecular mobility in the course of the milling procedure. Even though the liquid does not play the role of solvent, it can be incorporated into the crystal network to form solvates or not.

Herein we report the LAG method for synthesizing the tetrahydrate, depicted in Scheme 1, and anhydrate of the antidiabetic drug metformin and *N*,*N*′-(1,4-phenylene)dioxalamic acid. Their molecular and supramolecular structures were analyzed in the context of the interactions of metformin in biologic systems.

## 2. Materials and Methods

### 2.1. Materials and Crystal Synthesis

Metformin hydrochloride (HMetCl) was isolated from commercial sources: 10 tablets of 500 mg were ground and suspended in 150 mL of ethyl alcohol (96%), the mixture was boiled under stirring for 10 min or until complete dissolution. The hot solution was filtered; crystals of HMfCl were obtained after cooling to room temperature to obtain 2.5 g of a white solid, which was analyzed by IR, NMR and single crystal X-ray diffraction, corresponding to the crystal structure reported by Hariharan [7]. The *N,N′*-(1,4-phenylene)dioxalamic acid double hydrate (H_2_pOXA∙2W, W = water) was obtained as reported elsewhere [14]. Single crystals of *N,N′*-(1,4-phenylene)dioxalamate of di-(metformin diammonium) tetrahydrate (H_2_Mf(HpOXA)_2_∙4W) were harvested starting from 0.060 g (0.350 mmol) of HMfCl and 100 mg (0.347 mmol) H_2_pOXA**∙**2W which were grinded with a pestle in a mortar with the aid of few drops of water for 33–45 min until a homogeneous paste was formed. This mass was suspended in 40 mL of water and boiled under stirring until a clear solution appeared. The solution was left to stand at room temperature and after two days, 0.113 g (0.16 mmol) of beige crystals suitable for X-ray analysis were obtained in 92% yield. Similar results were obtained using 1:2 stoichiometric amounts of HMfCl (0.030 mg) to H_2_pOXA**∙**2W (100 mg) to obtain 0.120 g (0.17 mmol) of H_2_Mf(HpOXA)_2_∙4W in 98% yield. Microcrystalline powder of H_2_Mf(HpOXA)_2_ was obtained from single crystals of H_2_Mf(HpOXA)_2_∙4W after drying at 100 °C for 2 h in an air oven.

### 2.2. Instrumental

IR spectra were recorded neat at 25 °C using a Perkin Elmer Spectrum GX series with FT system spectrophotometer using the ATR device. ^13^C-CPMAS spectra were recorded on a Bruker Avance DPX-400 (101 MHz). The following conditions were applied: spectral width 30.242 kHz, acquisition time 33.8 ms, contact time 2000 ms, rotation rate 8 kHz, relaxation delay 5 s, and up to 256 scans for each spectrum were collected. Room temperature X-ray powder diffraction data were collected on a PAN Analytical X′Pert PRO diffractometer with Cu Kα1 radiation (λ = 1.5405 Å, 45 kV, 40 mA) or on a D8 Focus Bruker AXS instrument using Cu Kα1 radiation (λ = 1.542 Å, 35 kV, 25 mA). Texture analysis of H_2_Mf(HpOXA)_2_ was performed in an ASAP-2050 Xtended Pressure Sorption Analyzer of Micromeritics. Prior to measurement, vacuum sample activation was performed for 10 min to 150 °C, the measuring temperature was 75.15 K and it was maintained through a liquid nitrogen dewar. The gas used for the analysis was N_2_ (gas). For measurements, 40.8 mg of the activated mass sample were taken. DSC and TG measurements were performed in a Q2000 equipment and a Thermobalance Q5000 IR, respectively, of TA instruments. In both cases, approximately 3.0–5.0 mg of sample was used and a gradient of 5.00 °C/min from room temperature to 350 °C under air flux of 25 mL/min in an open (TG) or pin-holed panels (DSC).

### 2.3. X-ray Structure Determination

General crystallographic data for H_2_Mf(HpOXA)_2_**∙**4W has been deposited in the Cambridge Crystallographic Data Centre (CCDC) as supplementary publication number 1874280. Single crystal X-ray diffraction data were collected on an Agilent SuperNova (dual source) diffractometer using graphite-monochromatic Mo (λ = 0.71073 Å) Kα radiation; data collection, cell refinement and data reduction were accomplished using CrysAlisPro software [21]. The structure was solved by direct methods using the SHELXS program [22] of the WINGX package [23]. The final refinement was performed by full-matrix least-squares methods using the SHELXTL-2018/3 program [24]. The H atoms on C, N, and O were geometrically positioned and treated as riding atoms with: C–H 0.93−0.98 Å, Uiso(H) = 1.2 eq(C) for aromatic carbon atoms or 1.5 eq(C) for methyl carbon atoms; O–H = 0.82 Å, Uiso(H) = 1.5 eq(O); N–H = 0.86 Å, Uiso(H) = 1.2 eq(N). Platon [25] and Mercury [26] were used to prepare the material for publication.

Crystal Data for 2(C_10_H_7_N_2_O_6_)**∙**(C_4_H_13_N_5_)·4(H_2_O), (H_2_Mf(HpOXA)_2_**∙**4W), M = 705.61 g/mol: triclinic, space group *P*-1, a = 8.1357 (11), b = 13.8594 (18), c = 14.4846 (13) Å, α = 109.963 (10)°, β = 92.453 (9)°, γ = 95.863 (11)°, V = 1521.8 Å^3^, Z = 2, T = 293 K, μ(MoKα) = 0.71073 Å, Dcalc = 1.540 g/cm^3^, 10,558 reflections measured (3.0° ≤ 2θ ≤ 52.57°), 5941 unique (Rint = 0.027, Rsigma = 0.034), which were used in all calculations. The final R1 was 0.047 (I > 2σ (I)) and wR2 was 0.138 (all data).

## 3. Results and Discussion

### 3.1. Synthesis

Microcrystalline solid phase of *N,N′*-(1,4-phenylene)dioxalamate of di-(metformin diammonium) tetrahydrate (H_2_Mf(HpOXA)_2_∙4W) was synthesized using a water-assisted grinding procedure. Two stoichiometric proportions were used of HMfCl/H_2_pOXA: 1:1 and 1:2. In both cases, the reaction was monitored to completion by comparing the XRPD patterns of the mixtures to the pristine HMfCl, Figure 1. Single crystals of H_2_Mf(HpOXA)_2_**∙**4W organic salt were obtained by recrystallization from hot water of either of the two grinded mixtures. The reaction is quantitative based on the stoichiometry of the following reaction:2HMf^+^ + 2H_2_pOXA + 4H_2_O → H_2_Mf^2+^ + H_2_Mf(HpOXA)_2_**∙**4H_2_O

The proton of one carboxylic acid group of H_2_pOXA is transferred to the remaining basic nitrogen site of HMfCl molecule to form the H_2_Mf^2+^ dication and the monoanion of *N*,*N*′-(1,4-phenylene)dioxalamic acid (HpOXA^−^), which crystallize from the aqueous solution as H_2_Mf(HpOXA)_2_**∙**4W organic salt. Therefore, even when it can be obtained by a milling procedure in the solid state, the crystallization step from aqueous solution is required in order to eliminate the remaining H_2_MfCl_2_ byproduct.

Microcrystalline powder of H_2_Mf(HpOXA)_2_ was obtained from single crystals of H_2_Mf(HpOXA)_2_∙4W after drying at 100 °C for 2 h in an air oven.

### 3.2. The Molecular and Supramolecular Structure of H_2_Mf(HpOXA)_2_∙4W

The salt H_2_Mf(HpOXA)_2_**∙**4W crystallizes in the triclinic system, space group *P-*1 with one, two and four independent units of H_2_Mf^2+^, HpOXA^−^ and H_2_O in the asymmetric unit, respectively, and whose molecular structure is depicted in Figure 2.

The oxalyl fragments COCO are almost planar, with a mean O–C–C–O angle of 179.3(6)°. However, small differences can be noted between the oxalamic acid NCOCO_2_H and oxalamate fragments, NCOCO_2_^−^.The NCOCO_2_H endings are located slightly out of plane of the corresponding benzene ring, the maximum deviations from planarity are presented by N7C8O8C9O9O10H10 and N27C28O28C29O29O30H30 fragments with torsion angles of 16.08(5)° and 9.83(6)° from the C1–C6 and C21–C26 rings, respectively. Instead, the oxalamate counterparts NCOCO_2_^−^ are almost coplanar to the corresponding benzene ring: 3.50(6)° for N17C18O18C19O19O20 and 5.02(6)° for N37C38O38C239O39O40. It is worth noting that NCOCO_2_H and NCOCO_2_^−^ arms are in *syn* disposition between each other (*sp*-*sp* conformation). It is worth mention that this conformation has not been observed among organic cocrystals of H_2_pOXA but is commonly attained by coordination to metals [27]. The calculated conformational landscape of 1,4-phenylen dioxalyls predicts a very small difference in energy between planar *ap*-*sp* and *sp*-*sp* conformers of 0.26–0.28 kcal mol^−1^, in favor of the former, and an interconversion energy of only 4.80 kcal mol^−1^ [16]. Then, the *sp*-*sp* conformation exhibited by the HpOXA^−^ moiety is explained because of the stabilization given by hydrogen bonding with metformin that provides the energy to overcome the rotational barrier.

The C*(sp^2^)*–N*(sp^2^)* bond lengths of the metformin dication moiety range from 1.302(3) to 1.378(3) Å. In fact, the bond lengths of C52 and C54 with the terminal nitrogen atoms are shorter, whereas the corresponding bond lengths with the bridge N53 atom are longer, than those bonds observed in monocationic metformin (1.333–1.341 Å) [28]. These last bond lengths have values close to those observed in five-membered heterocycles involving pyrrolic nitrogen (≈1.37 Å) [29]. In agreement, the proposed delocalized structure is depicted in Figure 3. Selected bond lengths and torsion angles are listed in Table 1. In H_2_Mf(HpOXA)_2_**∙**4W, the two planar guanidinium halves N53C54N55N56 and N53C52N51N50, are twisted by 55.67(9)° and the NMe_2_ group is located opposite to the C(NH_2_)_2_ group (N56C54N53C52 torsion angle value of 148.8(2)°).

In the salt H_2_Mf(HpOXA)_2_**∙**4W, the metforminium moiety adopts the S backbone conformation, which has been reported as the most stable, owing to the decreased van der Waals repulsion, greater π-electron delocalization and intramolecular hydrogen bonding [9]. The structure of metformin in the H_2_Mf(HpOXA)_2_**∙**4W is very similar to that observed in the monohydrates of dicationic 1:1 salts of formula H_2_MfA**∙**H_2_O (A = oxalate, sulfate) [30], supporting that the anions slightly influence the structure of *N,N*-dimethylbiguanidinium moiety.

The H_2_Mf^2+^ and both HpOXA^−^ moieties are attached to each other through N53−H53···O38 and N50−H50B···O40, which lead to *R^2^_2_(9)* hydrogen bonding ring motif and N51−H51B···O18 hydrogen bonding interactions. Four water molecules, forming amide∙∙∙water interactions of bifurcated type in *R^1^_2_(6)* ring motif (Nn–Hn∙∙∙Om∙∙∙Hp—Cp; n = 7, 17; m = 3, 1; p = 6, 5), water∙∙∙carboxylate (O2–H2B∙∙∙O19^−^) and water∙∙∙water (O3–H3A∙∙∙O4) hydrogen bonding interactions, complete this basic repetition unit, Figure 4a. A duplex is formed by an inversion center of symmetry linked by N55−H55B···O28 hydrogen bonding, n→π * charge assisted interactions between O8···C54···O18, O8···N56, Figure 4b and several CO···CO interactions. The geometric parameters associated with these interactions are listed in Table 2. It is worth mention that the C···A distances (A = O, N) are smaller than the sum of the van der Waals radii of the involving atoms (r_VDW_ = 1.70 (C), 1.55 (N), 1.50 (O) Å) [31], and the C=O···C=O and C=O···N angles are in agreement with sheared parallel and perpendicular motifs, respectively [32]. Double strands are generated through acid∙∙∙carboxylate (On–Hn∙∙∙Om^−^; n =10, 30; m = 40, 20), Figure 5a, and N50−H50B···O40 hydrogen bonding, Figure 5b. Meanwhile, the second dimension is developed by water∙∙∙HpOXA^−^ interactions (On–Hn∙∙∙Om^−^; n = 2, 4; m = 39, 29), N37−H37···O1 and N55−H55A···O2, to form double layers within the (1 1 −1) plane, Figure 5b. A view along the direction perpendicular to this plane let us note two well-defined regions of metformin and water, Figure 5c. The distance between two HpOXA^−^ chains is longer in the metformin region than in the water region, in order to accommodate the NMe_2_ group, with mean values of 7.4(3) Å and 4.1(3) Å, respectively (distances measured between benzene ring edges). Finally, water molecules link the bilayers to develop the third dimension along the [0 1 1] direction (N27–H27···O2, N50–H50A···O4, N51–H51A···O4, O1–H1A∙∙∙O3, O1–H1B∙∙∙O19, O3–H3B∙∙∙O39, O4–H4A∙∙∙O8 and O4–H4A∙∙∙O10). The water molecules labelled as H_2_O1 (W1), H_2_O3 (W3) and H_2_O4 (W4) form an open cluster, but together with H_2_O2 (W2) are located in isolated pockets of the crystal lattice. The geometric parameters associated with intermolecular hydrogen bonding are listed in Table 3.

### 3.3. The Synthesis, Molecular and Supramolecular Structures of H_2_Mf(HpOXA)_2_ Anhydrate

Several techniques, such as IR and ^13^C-CPMAS spectroscopies, X-ray powder diffraction (XRPD), and thermal and BET-adsorption analyses, were performed in order to elucidate the molecular and supramolecular structure of the microcrystalline anhydrate of formula H_2_Mf(HpOXA)_2_.

The salt H_2_Mf(HpOXA)_2_∙4W exhibits a sequence of two weight losses at peak temperatures of 65 °C and 113 °C as endothermic processes, that correspond to the release of one (exp. 2.6%, calcd. 2.7%) and three water molecules (exp. 7.8%, calcd. 8.1%) per formula unit, respectively; Figure 6. The remaining solid is stable between 150 and 200 °C and decomposes at a peak temperature of 212–214 °C, releasing the equivalent to 53% of the initial mass; Figure 6. Thus, a new microcrystalline phase of composition H_2_Mf(HpOXA)_2_ was obtained after air drying at 100 °C, this anhydrate rehydrates into the original tetrahydrate under 100% RH at 40 °C. The XRPD patterns that confirm the crystallinity and change in the solid phases are shown in Figure 7.

A summary of the IR wavenumbers of H_2_Mf(HpOXA)_2_ tetrahydrate and anhydrate compared to HMfCl are listed in Table 4, and the corresponding IR spectra are shown in Figure 8. The νC=O stretching bands of the carboxylic acid and amide groups as well as the νC=N wavenumbers are red shifted in comparison to the starting H_2_pOXA∙2W and HMfCl as a result of hydrogen bonding interactions. As far as the carboxylate group is concerned, it appears at 1524 cm^−1^ in both H_2_Mf(HpOXA)_2_∙tetrahydrate and anhydrate, more red shifted than the reported value of 1540 cm^−1^ for the K_2_pOXA salt [14], which is in agreement with a highly delocalized structure. It is worth noting that dehydration considerably clears the spectral window between 3500 and 3000 cm^−1^ allowing the observation of the νN–H bands of the metformin moiety, which are observed in pairs in the IR spectrum of compound H_2_Mf(HpOXA)_2_, but the νN–H bands corresponding to the amide could not be assigned with certainty.

On the other hand, the ^13^C-CPMAS provides useful information regarding the molecular and supramolecular structure. The chemical shift data are listed in Table 5 and spectra are depicted in Figure 9. The chemical shift of the amide carbonyl (C8O) is sensitive to the change in the hydrogen bonding environment, it is spread over a 155–159 ppm range, in response to the hydrogen bonding scheme in the crystal lattice of H_2_Mf(HpOXA)_2_∙4W. Meanwhile, both COOH and COO^−^ groups appear as one signal slightly shifted by ∼2 ppm to high frequencies compared to the starting H_2_pOXA∙2W. This result is similar to that found in other carbonyls because of hydrogen bonding [33]. The *sp-sp* disposition between both oxalyl groups, in relation to the phenyl plane, and the presence of two independent molecules of HpOXA^−^ in the asymmetric unit, as well as hydrogen bonding, have the effect of spreading the CH signals of the benzene ring in the 121–118 ppm range. The loss of water molecules opens the spectral window from 123 to 117 ppm and shifts the amide carbonyl to low frequencies by ~2 ppm. The chemical shifts of the dicationic metformin moieties, in both tetrahydrate and anhydrate organic salts of (H_2_Mf)(HpOXA)_2_, are the same as those observed in the monocationic HMfCl, except that the value of the carbon atom in the C(NH_2_)NMe_2_ fragment is shifted by ~2 ppm towards high frequencies, which is in agreement with increased delocalization and positive charge (vide supra). The analysis of the vibrational spectra and ^13^C-CPMAS NMR data of H_2_Mf(HpOXA)_2_·4W and its anhydrate suggest that their structures, conformation and hydrogen bonding schemes of both organic salts are very similar between them, with the exception of those regions formerly occupied by water molecules. Rehydration experiments performed on the H_2_Mf(HpOXA)_2_ anhydrate, which reversibly rehydrates to the tetrahydrate, support these findings.

Rehydration experiments performed on the H_2_Mf(HpOXA)_2_∙anhydrate, which reversibly rehydrates to the tetrahydrate H_2_Mf(HpOXA)_2_**∙**4W, support the above findings. Furthermore, dehydration of H_2_Mf(HpOXA)_2_**·**4W can be associated with the role of water molecules in its crystal structure. As mentioned before, it occurred in two steps: one water molecule is lost at 65 °C followed by three more at 113 °C before decomposition. Judging by the number of hydrogen bonds and their strength, the water molecule labelled as W2 can be assigned to the first loss; it forms strong hydrogen bonds with amide and metformin NH in the crystal, as well as acting as a bridge between two HpOXA^−^ moieties of parallel chains; see Figure 5. Meanwhile, the water molecules labelled as W1, W3 and W4 form similar hydrogen bonds in number and nature as W2, but they are clustered together (vide supra), therefore leaving the crystal lattice at higher temperature. The removal of all water molecules from the isolated pockets of H_2_Mf(HpOXA)_2_**·**4W results in a stable framework given by the ionic interactions of the H_2_Mf^2+^ dication and two units of the HpOXA^−^ monoanion. This framework is stable and capable of rearranging in the presence of moisture, restoring the microcrystalline lattice of the original tetrahydrate. A similar behavior was observed in sitafloxacin hydrate [34]. The formation of microporous channels in the structure of H_2_Mf(HpOXA)_2_ was disregarded with a BET adsorption experiment, Figure 10a, which resulted in type III isotherm, characteristic of mesoporous solids [35], with pore size of ~16 nm, pore volume of 29 mm^3^ g^−1^ and surface area of 7.7 m^2^ g^−1^. The formation of large pores is in agreement with the ordered water detaching from the crystal structure, making available space along which water can diffuse during rehydration. In addition, the observed hysteresis loop, of type H3, indicates the presence of macropores and plate-like particles. Thus, the main role of the water in the crystal lattice of H_2_Mf(HpOXA)_2_**·**4W is proposed not only to compensate the excess of donors given by the metformin moiety, but also to provide the hydrogen bonding interactions to build the third dimension in the crystal lattice. The examination of the H_2_Mf(HpOXA)_2_**·**4W crystal lattice allows us to propose that most of the original hydrogen bonding interactions as well as n→π * charge-assisted interactions in the (0 1 1) plane are preserved after dehydration. Meanwhile, the (1 1 −1) plane is weakened after water loss, to form the H_2_Mf(HpOXA)_2_ mesoporous solid phase, Figure 10b.

### 3.4. The Nature and Structure of Dicationic Metfomin in H_2_Mf(HpOXA)_2_∙4W and H_2_Mf(HpOXA)_2_ Organic Salts

Metformin behaves as diprotic acid, whose pKa values are referred to here as pK_a1_ and pK_a2_, corresponding to H_2_Mf^2+^/HMf^+^ and HMf^+^/Mf acid-base conversions, respectively. The reported pK_a_ values are spread over a wide range depending upon the experimental method of measurement [36]. The pK_a1_ and pK_a2_ values are reported as being between 0.7–3.1 and 11.5–15.3 intervals, respectively. The formation of the organic salt H_2_Mf(HpOXA)_2_**∙**4W, vide supra, is explained by the proton transfer from H_2_pOXA, as the acid, to HMf^+^, as the base. The ΔpKa rule stablishes that ionized acid–base molecular complexes, in the solid phase, are observed exclusively for ΔpKa > 4 [37]. The ΔpKa is the difference between the pKa of the base and the pKa of the acid. Therefore, the ΔpKa rule lead us to estimate the pK_a1_ value for the first deprotonation event of H_2_pOXA in the −3.3 to −0.9 range, corresponding to a strong acid.

Furthermore, the protonation of the nitrogen bridge, in dicationic metformin, avoids the formation of the homodimer *R^2^_2_(8)*, raised by N–H∙∙∙N interactions. This motif is frequently observed in the crystal structures of the monocationic metformin salts but absent in all dicationic known salts of metformin [6] including the tetrachlorocuprate [38]. Instead, a hydrogen bonding heterodimer *R^a^_d_(n)* is formed in dicationic salts, with the participation of the anion group: *R^2^_2_(8)* in the oxalic acid [30] and nitrate [39], *R^2^_2_(9)* in squarate [40] and oxalamate monoacid (this work), *R^3^_3_(12)* motif in sulfate [30] and *R^2^_2_(16)* in sulfonatocalix [4,5]arenes [41]. The conformation in the form of an S backbone and the strongest acid on the nitrogen bridge seems to favor the hetero- over the homo-association. At this point, it is worth highlight the differences between metformin–carboxylate and metformin–oxalamate salts. The first forms the hydrogen bonding hetero-dimeric *R^2^_2_(8)* motif, which has been found, in the crystal network of guanidinium–carboxylate synthons [9]. This dominant motif is absent in the oxalamate salt H_2_Mf(HpOXA)_2_**∙**4W. Instead, the amide carbonyl of the HpOXA^−^ moiety participates in the formation of the observed *R^2^_2_(9)* motif and in the n→π * charge-assisted interactions of mono- and bifacial type, herein described (see Table 2). Even when weak in strength, the n→π * interaction, also named π-hole, has been found to be important for the stability of biomolecules and materials [42]. Because of the nature and bonding of the atoms involved, the carbonyl–guanidinium synthon CO∙∙∙CN^+^ is similar to CO···CO interactions [32], but charge assisted. This synthon, as well as H_2_O∙∙∙CN^+^ synthon, is present, even when not described in the original sources, in the crystal network of metformin dicationic salts of squarate and oxalate. The analogous synthon SO∙∙∙CN was identified in the crystal network of the acesulfame [43] and also in sulfonatocalix [4,5]arene metformin dicationic salts, in this last in the bifacial fashion SO∙∙∙CN^+^∙∙∙OS. Selected geometric parameters are listed in Table 6 to support the above discussion.

It has been revealed that the n→π * interaction can modulate the overall structural motifs even in the presence of strong hydrogen bonding interactions [44]. The results herein described for recurrent n→π * charge-assisted interactions of mono-and bifacial type, in dicationic metformin salts, support the existence of these interactions independently, and not just as a short contact imposed by the geometric constraint due to the hydrogen bonding patterns. This O(δ−)∙∙∙C(δ+) and O(δ−)∙∙∙N^+^ could be responsible of the interactions of metformin in biologic systems.

## 4. Conclusions

The H_2_Mf(HpOXA)_2_∙4W framework is given by ionic interactions of the H_2_Mf^2+^ dication and two units of the HpOXA^−^ monoanion. The inherent flexibility of both components is lost by the strong hydrogen bonding and n→π * charge-assisted interactions between both moieties, whereas the main role of the water molecules is to equilibrate the number of hydrogen bonding acceptors in the crystal network. The vacant pockets left behind after water removal are preserved in the H_2_Mf(HpOXA)_2_ anhydrate, whose crystal network is stable and capable to rearrange in the presence of moisture to restore the microcrystalline lattice of the original tetrahydrate under proper conditions.

The formation of the organic salt H_2_Mf(HpOXA)_2_**∙**4W implies the proton transfer from H_2_pOXA, as the acid, to HMf^+^, as the base. The first deprotonation event of H_2_pOXA was estimated in the −3.3 to −0.9 range, corresponding to a medium strength acid. The structure of metformin in the 1:2 salt H_2_Mf(HpOXA)_2_**∙**4W is very similar to that observed in the monohydrates of the two other known dicationic 1:1 salts of formula (H_2_Mf)A**∙**H_2_O (A = oxalate, sulfate). The conformation in the form of an S backbone and the strongest acid on the nitrogen bridge seems to favor the hetero-over the homo-association; therefore, dicationic metformin is prone to hydrogen bonding with anions to form hetero *R^a^_d_(n)* motifs. The *syn* conformation adopted by the *N*,*N*′-(1,4-phenylene)dioxalamic acid is uncommon among organic cocrystals, but is only attained by coordination to metals. The amide carbonyl of the HpOXA^−^ moiety participates in the formation of the observed *R^2^_2_(9)* motif and in the n→π * charge-assisted interactions of mono-and bifacial type. The n→π * charge-assisted interactions should be taken into account in modelling the interaction of metformin with biological targets more than affecting the metformin therapeutic effects.

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
