# Peer review of "Mechanochemical Synthesis and Structure of the Tetrahydrate and Mesoporous Anhydrous Metforminium(2+)-N,N′-1,4-Phenylenedioxalamic Acid (1:2) Salt: The Role of Hydrogen Bonding and n→π * Charge Assisted Interactions"

_pharmaceutics, 2020, doi:10.3390/pharmaceutics12100998_

Round 1
Reviewer 1 Report
The manuscript by Padilla-Martinez and co-Authors is an interesting and well written work.
There are only minor suggestions for what concerns the language. I would suggest the term (verb) “harvested” instead of “synthesized” at line 83, page 2. Furthermore there is an evident typo (suggested by the Authors’ native language) at line 231, page 8 where instead of “that” is present a “que”.
There are, however, some bigger concerns about the deposited .cif file. The checkcif procedure evidenced several C type alerts, not commented or explained by Authors. While the longer and shorter (respectively) bond distances for C-C bonds and H-bonds are not a real issue, due to the particular chemical environment, I’m going to list some of the most relevant flaws present in the cif.
- There is a very large scale factor (K = Mean[Fo**2] / Mean [Fc**2]) much bigger than 1 (2.352). It should be justified and commented.
- There are some missing reflections between Theta(Min) and sintheta/lambda=0.6
- There is one reflection with I(obs) << I(calc) i.e. (Fo**2 - Fc**2) / Sigma(Fo**2) < - 100.0.
These two last entries (2 and 3) should be investigated and explained.
At page 4, Figure 2, Authors reported the molecular structure of the hydrated co-crystal with the relative ellipsoid. While this choice is formally correct, I would have appreciated also the complete cell layout – with relative axes – and a table with the relevant cell parameters – angles and axes, together with the already reported SG.
What I think is, in my humble opinion, the most relevant issue in the manuscript is the lack of direct evidences, as strong as the one inferred by means of the single crystal X-ray diffraction analysis, for the study of the anhydrous structure. As the Authors performed a – good quality – XRPD analysis if the compound, I don’t understand why they don’t try to solve this structure. The lack of a direct comparison makes most of the arguments used to confirm the similarities – see page 10 lines 270-274 – less rigorous and somewhat hurried as compared to the previously used style and approach. This is also true for the BET analysis and its conclusions reported at page 11 (lines 296-300): the presence of “generic” large pores is in fact not only in agreement with water detach – as suggested by Authors. I expect some more detailed comments and experimental evidences about this topic.
Reviewer 2 Report
The authors successfully examined a new cocrystal salt of metformin and N,N’-(1,4-phenylene)dioxalamic acid. The structure determination was well carried out by single crystallographic and pXRD studies. IR, 13C-CPMAS, TG and BET adsorption studies were presented for understanding the solid states and water accessibility. They discussed the supramolecular association of the co-crystals, showing the selective formation of the monoanion HpOXA⁻ and the participation of n--->π* charge-assisted interaction, which are well written and interesting information.
Thus, I recommend the publication of this paper for the article of ‘Pharmaceutics’, especially for the special issue of ‘Controlled Crystallization of Active Pharmaceutical Ingredients’, after revision.
1. Please add the scheme of two molecular structures in the introduction to be understood the research by a wide range of readers.
2. Please review the numbering schemes of the crystal structure seriously! It's quite confusing in Figs 2~5 and Table 1. For example, there are two N52 in Figure 2, the numbering is different for carbon between Figs 2 and 3, and the two O39 in Fig. 5 is it correct? Please check the text, lines 165-206, and Table 1 (there are the same column). Such mistakes reduce the impression of the entire paper and make it hard to trust.
3. It is inappropriate to discuss the n ---> π* of O8-C54-O18 and O8-N56 from the distance of the crystal structure.
4. Please review the Figure number in the text; line 198, Figure 4(c) is probably Figure 5(c), line 228, Figure 5 is probably Figure 6. It is difficult to read because the order of Figure 5a and Figure 5b is reversed in the text.
5. In 'Materials and Methods', please add the equipment and details for adsorption experiment, which is only missing. In general, polar size is determined by inert gas. I recommend the author to describe the pretreatment temperature and time, the measured temperature and gas type, and the reproducibility of adsorption with respect to hysteresis in the methods (2) and/or discussion (3.3). The void space in the crystal when water is removed can be easily visualized and the values can be shown by Mercury. In addition, gas adsorption isotherms and BET analysis should be distinguished in the caption of Figure 10.
6. The authors could also consider to carry out a Hirschfield surface analysis to better quantify the number and type of non-covalent interactions.
Reviewer 3 Report
'Mechanochemical synthesis and structure of the tetrahydrate and mesoporous anhydrous metforminium(2+)-N,N’-1,4-phenylenedioxalamic acid cocrystal salt: the role of hydrogen bonding and n→π* charge assisted interactions' by Padilla-Martínez at all has described the detailed synthesis and analysis of the metformin solid form.
I strongly think that this solid form is not qualified to be called as salt cocrystal or ionic cocrystal, as there are no neutral species (apart from water) in the crystal lattice. Also, as described in the abstract it may not be the first crystal structure in the CSD/literature. However, it is not common to see such type of crystal structures for sure. I would call this type of crystals as double-salt rather than a salt cocrystal. Authors can go through the review written by Prof. Rogers at all in Green Chemistry with the title: Mixing ionic liquids – “simple mixtures” or “double salts”? (https://pubs.rsc.org/en/content/articlelanding/2014/gc/c3gc41389f#!divAbstract).
Please modify the paper accordingly.
The rest of the content and the analytical part of the manuscript have described adequately.
Author Response
Dear Reviewer,
Response: after checking the suggested reference some doubts raised, therefore we decided to call the title compound as organic salt or crystal salt.
Thank you
Reviewer 4 Report
This article by Padilla-Martinez and colleagues reported a new mechanochemical synthesized cocrystal salts of metformin and N,N’-(1,4-phenylene)dioxalamic acid. Authors provided a comprehensive characterization to investigate the new cocrystals structures and their physicochemical properties. They also provided decent analyses to consolidate the role of hydrogen bonding and n→π* charge-assisted interaction in the cocrystal synthesis and the metformin interactions in biologic systems. The authors therefore present a compelling study of the mechanochemical synthesized cocrystals. The manuscript is well written and adequately illustrated. Further perspectives can be discussed in the Conclusion section. For instance, would the intermolecular interaction observed in the cocrystals physiologically affect the therapeutic effects? There are several mislabeled atom numbers, especially in hydrogen bonding. Please go through the text, figures, and tables again to correct them.
Minor Concerns:
Figure 2: # 51-54 atom labeling of H2Mf were wrong.
Figure 5b: O39 in hydrogen bonding “N53-H53…O38” should be corrected as “O38”.
Table 3: stated hydrogen bonding N50-H50B…O40, N50-H50A…O4, N50-H50A…O9. While in text line 186 N50-H50A…O40, line 194 N50-H50B…O9, line 202 N50-H50B…O4. Please double check all numbering and correct them correspondingly.
Page 6, line 198: where is Figure 4(c) panel? Does it mean 5(c) here?
Page 8, line 228: figure 5 should be corrected to “figure 6”.
Figure 6: please state clearly in the figure of which curves stand for tga and dsc, respectively corresponding to the color.
Author Response
Further perspectives can be discussed in the Conclusion section. For instance, would the intermolecular interaction observed in the cocrystals physiologically affect the therapeutic effects?
Response: The following text was included in the Conclusion section (lines 366-367):
The n→p* charge-assisted interactions should be taken into account in modelling the interaction of metformin with biological targets more than affecting the metformin therapeutic effects.
Minor Concerns:
Figure 2: # 51-54 atom labeling of H2Mf were wrong.
Response: The labelling scheme was corrected. I apologize for this mistake.
Figure 5b: O39 in hydrogen bonding “N53-H53…O38” should be corrected as “O38”.
Response: Done, in addition this figure was modified in order to label only the atoms viewed in the front.
Table 3: stated hydrogen bonding N50-H50B…O40, N50-H50A…O4, N50-H50A…O9. While in text line 186 N50-H50A…O40, line 194 N50-H50B…O9, line 202 N50-H50B…O4. Please double check all numbering and correct them correspondingly.
Response: The paragraph was revised and the atom labelling modified according to Table 3.
Page 6, line 198: where is Figure 4(c) panel? Does it mean 5(c) here?
Response: This mistake was corrected.
Page 8, line 228: figure 5 should be corrected to “figure 6”.
Response: This mistake was corrected.
Figure 6: please state clearly in the figure of which curves stand for tga and dsc, respectively corresponding to the color.
Response: Figure 6. Tg, in black, and DSC, in blue, of H2Mf(HpOXA)2∙4W.
Round 2
Reviewer 1 Report
Authors performed most of the suggested changes, thus improving both the manuscript and the relative .cif file deposited at CCDC.
The paper is now suitable for publication in Pharmaceutics.
Reviewer 2 Report
The authors revised the text and graphics for reviewer comments and I would like to recommend the publication of the current form.
Reviewer 3 Report
thanks for the revision I am satisfied with the revised version of this manuscript.